# MetricFormer: A Unified Perspective of Correlation Exploring in Similarity Learning

**Jiexi Yan**[1,2]   **Erkun Yang**[2]   **Cheng Deng**[2*]  **Heng Huang**[3]
[1] School of Computer Science and Technology, Xidian University, Xi'an 710071, China
[2] School of Electronic Engineering, Xidian University, Xi'an 710071, China
[3] Electrical and Computer Engineering, University of Pittsburgh, PA 15261, USA
{jxyan1995, erkunyang, chdeng.xd, henghuanghh}@gmail.com

## Abstract

Similarity learning can be significantly advanced by informative relationships among different samples and features. The current methods try to excavate the multiple correlations in different aspects, but cannot integrate them into a unified framework. In this paper, we provide to consider the multiple correlations from a unified perspective and propose a new method called MetricFormer, which can effectively capture and model the multiple correlations with an elaborate metric transformer. In MetricFormer, the feature decoupling block is adopted to learn an ensemble of distinct and diverse features with different discriminative characteristics. After that, we apply the batch-wise correlation block into the batch dimension of each mini-batch to implicitly explore sample relationships. Finally, the feature-wise correlation block is performed to discover the intrinsic structural pattern of the ensemble of features and obtain the aggregated feature embedding for similarity measuring. With three kinds of transformer blocks, we can learn more representative features through the proposed MetricFormer. Moreover, our proposed method can be flexibly integrated with any metric learning framework. Extensive experiments on three widely-used datasets demonstrate the superiority of our proposed method over state-of-the-art methods.

## 1   Introduction

Exploring how to effectively learn semantic similarities among examples is an important problem in the field of machine learning and computer vision. Metric learning aims to transform the original data into an embedding space and learn a distance metric in this space to promote intra-class compactness and inter-class separability. In recent years, benefiting from the emerging advance of deep learning technique [20, 17], deep metric learning (DML) can capture more representative embeddings, and has been widely applied to various downstream tasks, including image retrieval [29, 28], face recognition [19, 36], and person re-identification [57].

The core idea of DML is to excavate the intrinsic structural relations of data, and utilize them to learn discriminative embeddings. As shown in Figure 1, such relations can be divided into two main categories: sample relationship and feature relationship. The former, *i.e.* sample relationship, includes *pairwise correlation* and *batch-wise correlation*. Most of existing supervised DML methods utilize *pairwise correlations* (*i.e.*, positive and negative pairs) to construct and utilize valid tuples of pairs through well-designed pair-based losses, *e.g.* triplet loss [36] and multi-similarity loss [46]. Due to training with random sampling, such methods suffer from heavily low convergence and model degradation. To discover the implicit global structure of the dataset, recent studies started to explore

---

*Corresponding Author.

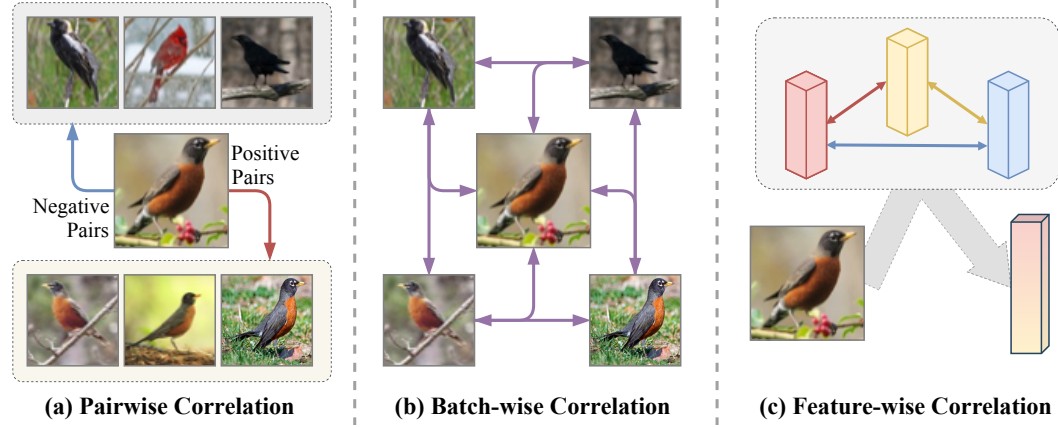

| (a) Pairwise Correlation | (b) Batch-wise Correlation | (c) Feature-wise Correlation |

Figure 1: An illustration of multiple intrinsic correlations among different samples and features. (a) According to the labels, we can naturally derive a series of positive and negative pairs, which is widely used in DML. (b) The relationships between different samples are more informative, and not limited to the samples from the same class. Samples in the same class have some invariant features, while similar classes share similar parts. (c) The ensemble of features also has inner structural relations between different sub-features, which helps the feature ensemble comprehensively represent samples.

the *batch-wise correlation* shown in Figure 1 (b) in the embedding space [18, 12, 1, 49, 37]. For example, IBC [12] adopt the message passing strategy to capture the overall structure of data in each mini-batch.

Considering that using only a single vector for similarity measurement cannot fully capture the semantics of inputs and exacerbates the unexplainability caused by the black-box nature of deep neural networks, recent studies [22, 29, 56] learn an ensemble of embeddings to jointly represent the input instance. Such ensemble-based method generally outperforms those that use a single feature vector since the ensemble encourages the learned embeddings to capture more comprehensive characteristics. However, simply concatenating the ensemble of sub-features ignores the structural relations among them, *i.e. feature-wise correlation* shown in Figure 1 (c), which is crucial to ensure each sub-feature modeling different and discriminative characteristics. For better performance and generalization ability, the feature-level correlation should also be considered during training.

In this paper, we provide a unified perspective to learn the multiple correlations and propose a metric transformer framework, named MetricFormer, which enables the information propagation along features of different samples within each mini-batch. Specifically, we introduce a correlation-aware transformer into the internal structure of deep metric networks, which contains a feature decoupling block, a feature-wise correlation block, and a batch-wise correlation block. We adopt the feature decoupling block with several trainable query vectors to learn a series of sub-features with different discriminative characteristics. Meanwhile, the feature-wise correlation block is performed to excavate the intrinsic relations among sub-features, which can ensure the learned ensemble of features is representative and sufficient for similarity measurement. Furthermore, to implicitly capture and model the sample relationships, we apply the batch-wise correlation block into the batch dimension of each mini-batch with the learned ensemble of features. Moreover, a local embedding module is proposed to focus on more informative neighborhood relationships without redundancy. Guided by the sparse graph, we can encode the local neighborhood relationships of each sample into the transformer feature.

Our work delivers the following contributions:

- We propose to simultaneously explore the multiple correlations among data from a unified learning perspective.

- We provide a simple yet effective method termed as MetricFormer to implicitly capture three levels of correlations in an end-to-end model.

- We achieve significant performance gains against the state-of-the-art methods on three widely-used benchmark datasets.

## 2 Preliminaries

### 2.1 Problem Formulation of DML

Given the training dataset $\mathcal{D} = \{(\mathbf{x}_i, y_i)\}_{i=1}^N$ where $y_i$ is the corresponding label of the sample $\mathbf{x}_i$, DML methods aims to learn a feature mapping $f : \mathcal{X} \mapsto \mathcal{Z} \in \mathbb{R}^D$ with deep neural networks, which maps input samples to a $D$-dimensional embedding space $\mathcal{Z}$. With the learned projection $f : \mathbf{z}_i = f(\mathbf{x}_i)$, we can effectively measure the similarity between any two samples $\mathbf{x}_i$ and $\mathbf{x}_j$ using Euclidean distance

$$d_{i,j} = \|\mathbf{z}_i - \mathbf{z}_j\|_2, \tag{1}$$

or cosine similarity

$$s_{i,j} = \frac{\mathbf{z}_j^\top \mathbf{z}_j}{\|\mathbf{z}_i\|_2^2 \|\mathbf{z}_j\|_2^2}. \tag{2}$$

To improve the ability of similarity measurement, a majority of DML studies develop various metric loss functions $\mathcal{L}_M$, which can impose a discriminative constraint on the feature embeddings.

### 2.2 Revisiting Transformer Encoder

Transformer architecture has achieved great success in the field of natural language processing and computer vision and brings novel paradigms for several fundamental tasks (*e.g.* translation and segmentation). The overall transformer model usually consists of a stack of multiple transformer encoder blocks. The transformer encoder contains a multi-head self-attention layer (MHSA) and a feed-forward network (FFN), which are followed by a LayerNorm (LN), respectively. Given a sequence of input features $\mathbf{X} \in \mathbb{R}^{M \times D}$ with the length of the sequence $M$ and the dimension of the input features $D$, we can derive the output of the transformer encoder as follows,

$$\begin{aligned} \bar{\mathbf{Y}}_l &= \mathrm{LN}\left(\mathrm{MHSA}(\mathbf{Z}_{l-1}) + \mathbf{Y}_{l-1}\right), \\ \mathbf{Y}_l &= \mathrm{LN}\left(\mathrm{FFN}(\bar{\mathbf{Z}}_l) + \bar{\mathbf{Z}}_l\right), \end{aligned} \tag{3}$$

where $l \geq 1$ is the index of layers in the transformer encoder, and $\mathbf{Z}_0 = \mathbf{X}$.

The self-attention mechanism is widely-used to capture and model the relationships from the channel and spatial dimensions. With $\mathbf{Q}, \mathbf{K}, \mathbf{V} \in \mathbb{R}^{M \times D}$ as the query, key, and value learned from the same input, we can calculate the output $\mathbf{Z}$ for the self-attention module:

$$\mathbf{Y} = \mathrm{softmax}\left(\frac{\mathbf{Q}\mathbf{K}^\top}{\sqrt{D}}\right)\mathbf{V}. \tag{4}$$

And then, the representations from different heads are concatenated.

## 3 The Proposed Method

In this section, we first introduce the overall motivation and framework of the proposed Metric Transformer, and then describe the three transformer blocks, respectively.

### 3.1 Metric Transformer

Since the relationships among samples and features are both various and complex, how to effectively discover the intrinsic structure of data and excavate more representative information from the training data is an essential problem in DML. Most of existing methods can only utilize the pair-based similarity, which neglects the batch-wise correlation and feature-wise correlation. Recently, some studies are proposed to mine such rich relations. However, they can only learn the batch-wise or feature-wise correlations individually, and the multiple correlations cannot be automatically learned in a unified end-to-end DML model.

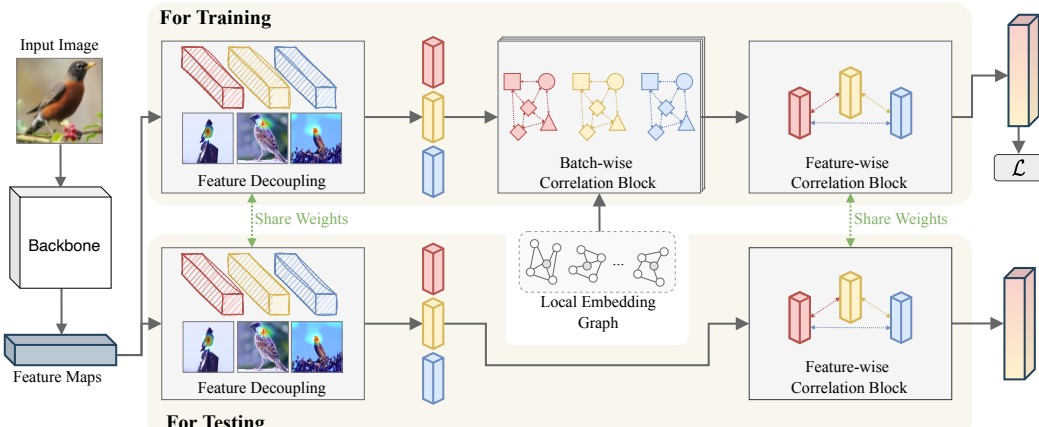

Figure 2: An illustration of the proposed MetricFormer framework. Our MetricFormer contains one feature decoupling block, one feature-wise correlation block, and $T$ batch-wise correlation blocks. After obtaining the feature maps generated by a backbone network, we can derive the ensemble of sub-features to represent the input samples from different aspects in the feature decoupling block. And then, the batch-wise correlation block and feature-wise correlation block with the same architecture employ the cross-attention mechanism from different dimensions to explore the relationships among samples and features. To tackle the inconsistency issue, we utilize the consistency preserving strategy during training.

We introduce a metric transformer (MetricFormer) to enable DML model itself with the ability to learn the sample relationships in each mini-batch and the structural relationships among the ensemble of sub-features. As shown in Figure 2, our MetricFormer consists of one feature decoupling block, one feature-wise correlation block, and $T$ batch-wise correlation block. Specifically, we adopt a backbone network to extract the feature maps from input samples. After that, a feature decoupling block is performed to split the feature maps into several sub-features with different discriminative characteristics. And then, the batch-wise correlation block is employed to model the relationships among different samples by utilizing the cross-attention mechanism in the transformer. Finally, we use the feature-wise correlation block to discover the structural pattern in the feature ensembles and obtain the aggregated features guided by the learned feature-wise relations.

**Consistency Preserving Strategy** Considering the inconsistency of input data structure between training and testing as the mini-batch of data is not available in testing, we apply a consistency preserving strategy, which forces the out features learned with the batch-wise block to be consistent with the ones learned without the batch-wise block. By doing this, we can remove the batch-wise correlation block during the test, while still benefiting from the sample relationship learning.

### 3.2 Feature Decoupling

Compared with learning the single embedding vector, employing an ensemble of embeddings with sufficient diversity can provide different characteristics of the input samples and improve the performance of DML. To achieve this, we exploit the self-attention mechanism with several trainable queries to automatically split the feature map into an ensemble of several distinct and diverse sub-features. The learned ensemble of features can comprehensively characterize the input sample from different aspects.

Specifically, we apply a backbone network to extract original feature maps $\mathcal{J} \in \mathbb{R}^{H \times W \times C}$, where $H, W, C$ denotes the height, width, and the number of channels, respectively. After that, we use two different linear layers $g_1(\cdot)$ and $g_2(\cdot)$ to process the feature map tensor $\mathcal{J}$ and derive the corresponding

key and value matrices in the feature decoupling block as follows:

$$\mathbf{K} = g_1(\boldsymbol{\mathcal{J}}) \in \mathbb{R}^{HW \times C_K}, \tag{5}$$

$$\mathbf{V} = g_2(\boldsymbol{\mathcal{J}}) \in \mathbb{R}^{HW \times d}, \tag{6}$$

where $C_K$ denotes the numbers of feature maps in the key matrix, and $d$ is the embedding dimension of the learned sub-features. Here, we set $d = [D/k]$ according to the number of sub-features $k$ and the final embedding dimension $D$.

To automatically learn the representative ensemble of features with the self-attention mechanism, we introduce a set of trainable query vectors $\mathbf{Q}_1 = [\mathbf{q}_1; \mathbf{q}_2; \cdots; \mathbf{q}_k] \in \mathbb{R}^{k \times C_K}$, where $k$ is the number of sub-features. These query vectors are randomly initialized and adaptively updated along with the other parameters in our model. To do so, we can calculate the attention score matrix as

$$\mathbf{A}_S = \text{softmax}\left(\frac{\mathbf{Q}\mathbf{K}^\top}{\sqrt{C_K}}\right) \in \mathbb{R}^{k \times HW}, \tag{7}$$

which measures the importance of spatial positions to guide the learned sub-features to identify different spatial positions. To this end, we can obtain the ensemble of features

$$\mathbf{Y} = \mathbf{A}_S \mathbf{V} \in \mathbb{R}^{k \times d}. \tag{8}$$

Different rows of $\mathbf{Y}$ represent the learned sub-features with different characteristics. Moreover, we employ a diversity loss to encourage the diversity between every two sub-features as follows:

$$\mathcal{L}_{div} = \log\left(1 + \exp(\alpha(s_{i,j} - m)\beta_0)\right), \tag{9}$$

where $s_{i,j}$ is the cosine similarity between two sub-features $\mathbf{Z}_i^0$ and $\mathbf{Z}_j^0$ of the same sample. And the metric loss is applied to each sub-feature of training samples separately to ensure that the learned ensemble of features can effectively represent the inputs from different aspects.

### 3.3 Multiple Correlations Exploration

The feature decoupling block enables the spatial information propagation and learns the ensemble of sub-features with discriminative characteristics from different aspects. With the learned ensembles of features, we can further explore the information propagation among different samples within each training mini-batch and different sub-features within each ensemble. Both of them can significantly improve the performance of DML.

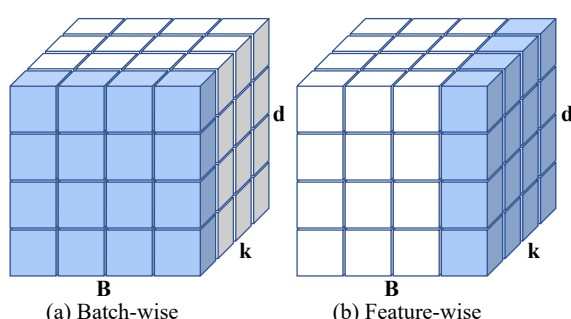

(a) Batch-wise      (b) Feature-wise

Figure 3: An illustration of the attention mechanisms on batch and feature dimensions. The batch-wise correlation block treats a mini-batch of samples as a sequence, while the feature-wise correlation block regards an ensemble of sub-features as a sequence.

To effectively capture the multiple correlations, we design two transformer encoder blocks with the same architecture, *i.e.* batch-wise correlation block and feature-wise correlation block. As shown in Figure 3, if we treat each sample in the mini-batch as a node of a sequence, the batch-wise correlation block can implicitly mine the relationships among samples in the mini-batch.

While if we regard each sub-feature of the sample as a node of a sequence, the structural relations among the sub-features through the feature-wise block can be explored. Considering a set of features $\boldsymbol{\mathcal{F}} \in \mathbb{R}^{B \times k \times d}$ in a mini-batch, we can calculate the corresponding query, key, and value tensors $\boldsymbol{\mathcal{Q}}, \boldsymbol{\mathcal{K}}, \boldsymbol{\mathcal{V}} \in \mathbb{R}^{B \times k \times d}$. After that, we apply the attention mechanism from different dimensions.

**Batch-wise Correlation Block**    To explore the sample relationships in each mini-batch, we generalize the attention mechanism into the batch dimension as follows:

$$\mathbf{A}_B = \text{softmax}\left(\frac{\boldsymbol{\mathcal{Q}}_{:i:}\boldsymbol{\mathcal{K}}_{:i:}^\top}{\sqrt{d}}\right) \in \mathbb{R}^{B \times B}, \tag{10}$$

where $\mathcal{Q}_{:i:}, \mathcal{K}_{:i:}, \mathcal{V}_{:i:} \in \mathbb{R}^{B \times d}$. The attention score matrix $\mathbf{A}_B$ can measure the sample relationship in this mini-batch. To further enhance the local manifold embedding, we introduce a local embedding graph $\mathbf{G}$ defined as

$$\mathbf{G}_{i,j} = \begin{cases} s_{i,j} & \text{if } \mathbf{x}_i \in \mathcal{N}(\mathbf{x}_i, n) \text{ and } s_{i,j} \geq \epsilon, \\ 0 & \text{otherwise,} \end{cases} \tag{11}$$

where $\mathcal{N}(\mathbf{x}_i, n)$ denotes the set of top-$n$ samples of $\mathbf{x}_i$ according to the rank list $\{s_{i,j}\}_{i=1}^{B}$, and $\epsilon$ is the threshold. And then, the attention score matrix can be updated as follows:

$$\bar{\mathbf{A}}_B = \mathbf{G} \odot \mathbf{A}_B, \tag{12}$$

where $\odot$ denotes the element-wise product operation. To this end, we can derive the output embeddings:

$$\begin{aligned} \mathcal{Y}_{:i:} &= \bar{\mathbf{A}}_B \mathcal{V}_{:i:}, \\ \mathcal{Y} &= \text{concat}(\mathcal{Y}_{:1:}, \mathcal{Y}_{:2:}, \cdots, \mathcal{Y}_{:k:}). \end{aligned} \tag{13}$$

**Feature-wise Correlation Block**    To excavate the structural relations among different sub-features, we apply the attention mechanism from the feature dimension. For each sample $\mathbf{x}_i$ ($i = 1, 2, \cdots, B$), we treat its ensemble of sub-features as a sequence, *i.e.*, we have $B$ sequences with the length of $k$. Specifically, the output embeddings are learned according to the query, key and value matrices $\mathcal{Q}_{i::}, \mathcal{K}_{i::}, \mathcal{V}_{i::} \in \mathbb{R}^{k \times d}$ in the position of the $i$-th sample:

$$\mathcal{Y}_{i::} = \text{softmax}\left(\frac{\mathcal{Q}_{i::}\mathcal{K}_{i::}^{\top}}{\sqrt{d}}\right)\mathcal{V}_{i::} \in \mathbb{R}^{k \times d}. \tag{14}$$

## 4  Related Work

**Deep Metric Learning**    Recently, deep metric learning has been well studied and achieved great advances [6, 36, 27, 47, 3, 4, 54, 53]. The majority of works focus on designing discriminative metric loss functions [14, 28, 46, 39, 21, 31, 38] and hard mining strategies [11, 10, 15, 48, 52, 50, 9] for training more effectively. Recently, some methods consider to learn an ensemble of embeddings to capture more characteristics of inputs from different aspects [22, 52, 35, 26, 56]. Nevertheless, the rich relationships among samples and embeddings are usually ignored in the existing methods.

**Transformer**    In recent years, based on multi-head self-attention mechanism [42], transformers, represented by BERT [7] and ViT [8], develop rapidly and have been widely used in the field of natural language processing and computer vision. Visual transformer [8, 25] has gradually become a new backbone for computer vision tasks, including classification [8, 25], detection [2], segmentation [55] and representation learning [5, 16].

## 5  Experiments

In this section, we evaluate the effectiveness of our method on three widely-used datasets on image clustering and retrieval tasks.

### 5.1  Experimental Settings

**Datasets**    To evaluate the performance of the proposed MetricFormer, we conduct the experiments on three widely-used benchmarks CUB-200-2011 (CUB) [43], Cars196 [24] and Stanford Online Products (SOP) [28]. Following the standard

Table 1: Statistics of three datasets.

| Datasets | Training | | Testing | |
|---|---|---|---|---|
| | # classes | # Samples | # classes | # Samples |
| CUB | 100 | 58,64 | 100 | 5,924 |
| Cars196 | 98 | 8,054 | 98 | 8,131 |
| SOP | 11,318 | 59,551 | 11,316 | 60,502 |

protocol [38, 28], we split the training and testing sets without intersection. As shown in Table 1, CUB-200-2011 contains 200 bird species of $11,788$ images. The first 100 categories of 5,864 images are used as the training set and the remaining 5,924 images from the other 100 categories are used for

testing. Cars196 includes 16,185 images from 196 car categories. The first 98 classes (8,054 images) are used for training and the rest 98 classes (8,131 images) for testing. Standard Online Products contains 120,035 images from 22,634 products. We use 59,551 images of the first 11,318 products for training and 60,502 images of the remaining 11,316 categories for testing.

**Evaluate Metrics**    Following the existing works [38, 28, 44], we conduct experiments on both image retrieval and clustering tasks. For the retrieval task, we calculate Recall@K to evaluate the retrieval performance. Recall@K measures the percentage of well-separated samples acknowledged if we can find at least one corrected retrieved sample in the $n$ nearest neighbors. For the clustering task, we adopt the Normalized Mutual Information (NMI) based on the K-means algorithm as the evaluation metric.

**Baselines**    We apply the proposed MetricFormer to the margin loss [48] and the ProxyAnchor loss [21] for comparison. We also compare our method with other state-of-the-art methods including pair-based methods, N-Pair [38], HTL [13], Angular [44], MS [46], Circle [39], proxy-based methods SoftTriple [31], ProxyGML [58], and ensemble methods HDC [52], A-BIER [29], ABE [22], DiVA [26], DRML [56].

**Implementation Details**    We implement all the following experiments using PyTorch [30] on an NVIDIA RTX 1080ti GPU, and the ADAM optimizer [23] with a learning rate of 0.0001 is adopted to train the model. We use ResNet-50 [17] as backbone model pre-trained on ImageNet dataset [34] for fair comparisons. For data augmentation, we use random cropping and horizontal mirroring. We set the number of sub-features $k$ to 3, and the embedding size of each sub-feature is fixed to 170. For each iteration, we fix the batch size to 112. For the margin loss [48], we set the margin factors $\alpha$ and $\beta$ to 1.2 and 0.2, respectively. For the ProxyAnchor loss [21], we set the temperature $\alpha = 16$, positive margin $\gamma_{pos} = 1.8$, and negative margin $\gamma_{neg} = 2.2$.

## 5.2    Experimental Results

**Quantitative Results**    We evaluate the performance of our method on three benchmarks CUB-200-2011 [43], Cars196 [24] and Stanford Online Products (SOP) [28]. The experimental results are shown in Table 2. Note that MetricFormer-PA and MetricFormer-M represent our method combined with ProxyAnchor and Margin loss, respectively. The best results are marked in bold red. We highlight our superior results over the associated methods without MetricFormer in bold black.

We can easily observe that our MetricFormer can improve the performance of the original DML methods by a significant margin. Compared with the original methods, our method can mine more informative relations from different aspects, which is useful for similarity learning. Moreover, our method outperforms the state-of-the-art methods on all three datasets. These empirical results support our proposal that our MetricFormer can discover the relationships among samples and features, which are ignored by most of the existing DML methods. Within the multiple relations, we can capture the characteristics of samples more comprehensively.

**Qualitative Results**    To further demonstrate the performance of our method, we give the visualization results of our MetricFormer on the CUB-200-2011 dataset. As shown in Figure 5, the decoupled sub-features focus on the body, neck, and head of birds respectively, which demonstrates that the learned ensemble of features can capture and model different characteristics. The trainable queries in the feature decoupling block can be regarded as pattern detectors and large weights are generated once a particular pattern is detected.

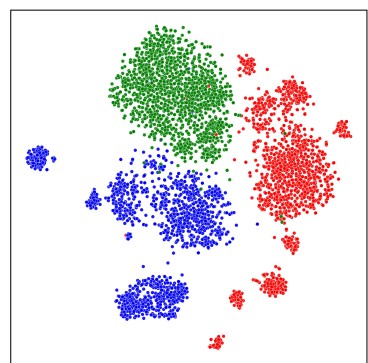

Moreover, we visualize the sub-features for all the samples on the CUB-200-2011 dataset using t-SNE [41] in Figure 5. We use different colors to represent the different sub-features in a 2-dimensional space. We observe that the learned sub-features are very diverse in the feature space. These visualization results support our proposal that our MetricFormer can learn

Figure 4: The t-SNE visualization results of the learned feature ensemble on the CUB-200-2011 dataset.

the ensemble of distinct and diverse features with different discriminative characteristics.

Table 2: Experimental results (%) of MetricFormer in comparison with the state-of-the-art methods on the testing sets of CUB-200-2011, Cars196 and Stanford Online Products. $n$-G/BN/R indicates the model setting where $n$ is the embedding size, and G,BN,R denote GoogleNet [40], Inception with Batch Normalization [20], and ResNet-50 [17], respectively.

| Datasets ↦ | | CUB | | | Cars196 | | | SOP | | |
|---|---|---|---|---|---|---|---|---|---|---|
| Methods ↓ | Setting ↓ | R@1 | R@2 | NMI | R@1 | R@2 | NMI | R@1 | R@10 | NMI |
| N-Pair [38] | 512G | 50.1 | 63.3 | 60.4 | 71.1 | 79.7 | 64.0 | 67.7 | 83.8 | 88.1 |
| Angular [44] | 512G | 53.1 | 65.0 | 61.0 | 71.3 | 80.7 | 62.4 | 67.9 | 83.2 | 87.8 |
| RLL-H [45] | 512BN | 57.4 | 69.7 | 63.6 | 74.0 | 83.6 | 65.4 | 76.1 | 89.1 | 89.7 |
| HTL [13] | 512BN | 57.1 | 68.8 | - | 81.4 | 88.0 | - | 74.8 | 88.3 | - |
| ProxyGML [58] | 512BN | 66.6 | 77.6 | 69.8 | 85.5 | 91.8 | 72.4 | 78.0 | 90.6 | 90.2 |
| SoftTriple [31] | 512BN | 65.4 | 76.4 | 69.3 | 84.5 | 90.7 | 70.1 | 78.3 | 90.3 | 92.0 |
| MS [46] | 512BN | 65.7 | 77.0 | - | 84.1 | 90.4 | - | 78.2 | 90.5 | - |
| GroupLoss[12] | 512BN | 65.5 | 77.0 | 69.0 | 85.6 | 91.2 | 72.7 | 75.1 | 87.5 | 90.8 |
| A-BIER [29] | 512R | 57.5 | 68.7 | - | 82.0 | 89.0 | - | 74.2 | 86.9 | - |
| ABE [22] | 512R | 60.6 | 71.5 | - | 85.2 | 90.5 | - | 76.3 | 88.4 | - |
| MIC[32] | 128R | 66.1 | 76.8 | 69.7 | 82.6 | 89.1 | 68.4 | 77.2 | 89.4 | 90.0 |
| TML [51] | 512R | 62.5 | 73.9 | - | 86.3 | 92.3 | - | 78.0 | 91.2 | - |
| DiVA [26] | 512R | 69.2 | 79.3 | 71.4 | 87.6 | 92.9 | 72.2 | 79.6 | 91.2 | 90.6 |
| PADS [33] | 128R | 67.3 | 78.0 | 69.9 | 83.5 | 89.7 | 68.8 | 76.5 | 89.0 | 89.9 |
| CircleLoss [39] | 512R | 66.7 | 77.4 | - | 83.4 | 89.8 | - | 78.3 | 90.5 | - |
| IBC [37] | 512R | 70.3 | 80.3 | 74.0 | 88.1 | 93.3 | 74.8 | 81.4 | 91.4 | 92.6 |
| DIML [54] | 512R | 68.2 | - | | 87.0 | - | | 79.3 | - | |
| DRML [56] | 512R | 68.7 | 78.6 | 69.3 | 86.9 | 92.1 | 72.1 | 79.9 | 90.7 | 90.1 |
| AVSL [53] | 512R | 71.9 | 81.7 | - | 91.5 | 95.0 | - | 79.6 | 91.4 | - |
| Margin[48] | 512R | 63.1 | 74.4 | 66.7 | 79.9 | 87.5 | 64.1 | 78.3 | 82.7 | 90.2 |
| MetricFormer-M | 512R | **69.2** | **79.6** | **71.0** | **89.5** | **93.8** | **73.1** | **79.8** | **83.6** | **92.0** |
| ProxyAnchor [21] | 512R | 69.7 | 80.0 | 72.3 | 87.7 | 93.0 | 75.7 | 78.4 | 90.5 | 91.0 |
| MetricFormer-PA | 512R | **74.4** | **83.7** | **75.4** | **91.8** | **95.4** | **76.2** | **82.2** | **92.6** | **92.7** |

## 5.3 Ablation Study

**Ablation Study of Different Components** We conduct experiments with the margin loss to analyze the effectiveness of different blocks in our MetricFormer. **Margin** denotes the baseline method of using the margin loss. **+FDB**, **+BCB**, **+FCB** are short for 'feature decoupling block', 'batch-wise correlation block', and 'feature-wise correlation block', respectively. Table 3 shows the experimental results on the CUB-200-2011 dataset. It is clear that our Metric-

Table 3: Ablation study with different model settings.

| Method | R@1 | R@2 | NMI |
|---|---|---|---|
| Margin | 63.1 | 74.4 | 66.7 |
| Margin + FDB | 66.8 | 76.5 | 67.7 |
| Margin + FDB + BCB | 68.7 | 79.3 | 70.8 |
| Margin + FDB + FCB | 68.4 | 79.1 | 70.6 |
| **Margin + MetricFormer** | **69.2** | **79.6** | **71.0** |

Former achieves better performance than all the compared counterparts and all of the proposed blocks contribute to the overall improvement, which demonstrates that exploring and capturing informative

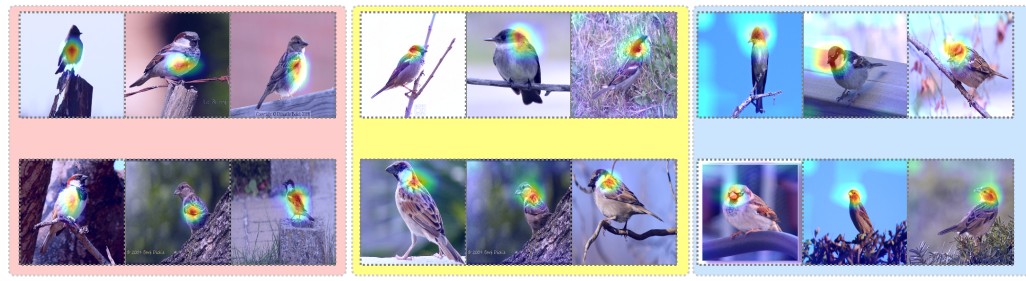

| Sub-feature 1 | Sub-feature 2 | Sub-feature 3 |

Figure 5: Attention map visualization results on the CUB-200-2011 dataset. The highlighted regions indicate the spatial locations that different sub-features are detecting. The three sub-features learned by our MetricFormer capture different characteristics of the bird images regardless of the spatial locations of these characteristics. The first sub-feature focuses on the body of birds, the second sub-feature focuses on the neck of birds, and the third sub-feature focuses on the head of birds.

relations is crucial for similarity learning. The relationships between samples and sub-features are both useful for similarity measurement.

**Influence of Different Number of Sub-features** We employ $k$ individual features to describe each input sample from different aspects. We conduct experiments with the margin loss (*i.e.* MetricFormer-M) to analyze the influence of the number of individual features. Table 4 shows the experimental results on the CUB-200-2011 dataset. The value of $k$ ranges from 2, 3 to 4, rendering the size of each sub-feature to 256, 170, and 128, respectively. We can observe that our method achieves the best performance when $k = 3$, and using a larger $k$ slightly harms the performance since each sub-feature captures less information with a larger $k$.

**Influence of Different Number of Batch-wise Correlation Blocks** $T$ We conduct experiments with the margin loss on the CUB-200-2011 dataset to analyze the effect of the number of batch-wise correlation blocks. According to the experimental results shown in Table 4, we can see that exploiting more batch-wise blocks can improve the model performance. Considering the amount of model parameters, we apply three batch-wise block in our MetricFormer, *i.e.* $T = 3$.

Table 4: Experimental results (%) of our MetricFormer with different number of the sub-features $k$ and different number of the batch-wise blocks $T$.

| $T = 3$ | | | | | | | | | $k = 3$ | | | | | | | | |
|---|---|---|---|---|---|---|---|---|---|---|---|---|---|---|---|---|---|
| $k = 2$ | | | $k = 3$ | | | $k = 4$ | | | $T = 1$ | | | $T = 2$ | | | $T = 3$ | | |
| R@1 | R@2 | NMI | R@1 | R@2 | NMI | R@1 | R@2 | NMI | R@1 | R@10 | NMI | R@1 | R@2 | NMI | R@1 | R@2 | NMI |
| 65.8 | 76.2 | 68.9 | 69.2 | 79.6 | 71.0 | 68.5 | 78.9 | 70.9 | 67.7 | 78.1 | 70.2 | 68.4 | 79.0 | 70.8 | 69.2 | 79.6 | 71.0 |

## 6 Conclusion

In this paper, we propose a metric transformer (MetricFormer) framework to explore multiple correlations in a unified perspective. In our MetricFormer, we take the pairwise relationship, the sample relationship among each mini-batch, and the structural relationship among the ensemble of features into consideration through the feature decoupling block, the batch-wise correlation block, and the feature-wise correlation block respectively. To this end, we can the ensemble of distinct and diverse features with different and discriminative characteristics. Moreover, our proposed method can be flexibly integrated with any metric learning method. We have conducted extensive experiments on three commonly used datasets which have demonstrated that our method can effectively boost the performance of the existing DML methods.

## Acknowledgment

Our work was supported in part by the National Natural Science Foundation of China under Grant 62132016, Grant 62171343, and Grant 62071361, in part by Key Research and Development Program of Shaanxi under Grant 2021ZDLGY01-03, and in part by the Fundamental Research Funds for the Central Universities ZDRC2102. Erkun Yang was supported in part by the National Natural Science Foundation of China under Grant 62202365, Guangdong Basic and Applied Basic Research Foundation (2021A1515110026), and Natural Science Basic Research Program of Shaanxi (program No.2022JQ-608).

## Broader Impact

**Limitation**    Due to the usage of transformer, the parameters of the model are increased, which leads to more computation cost and training times.

**Negative Impact**    The proposed method predicts content based on learned statistics of the training dataset and as such will reflect biases in those data, including ones with negative societal impacts. This issue warrants further research and consideration.

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

# A    Appendix

Optionally include extra information (complete proofs, additional experiments and plots) in the appendix. This section will often be part of the supplemental material.

