# Supplementary Material for " MetricFormer: A Unified Perspective of Correlation Exploring in Similarity Learning "

## A    Qualitative Results

To further demonstrate the performance of our method, we present some qualitative results in Figure 1. We can easily observe that the embeddings of our method can accurately retrieve the similar instances under various challenges, including pose variantion an background clutter in the CUB-200-2011 dataset.

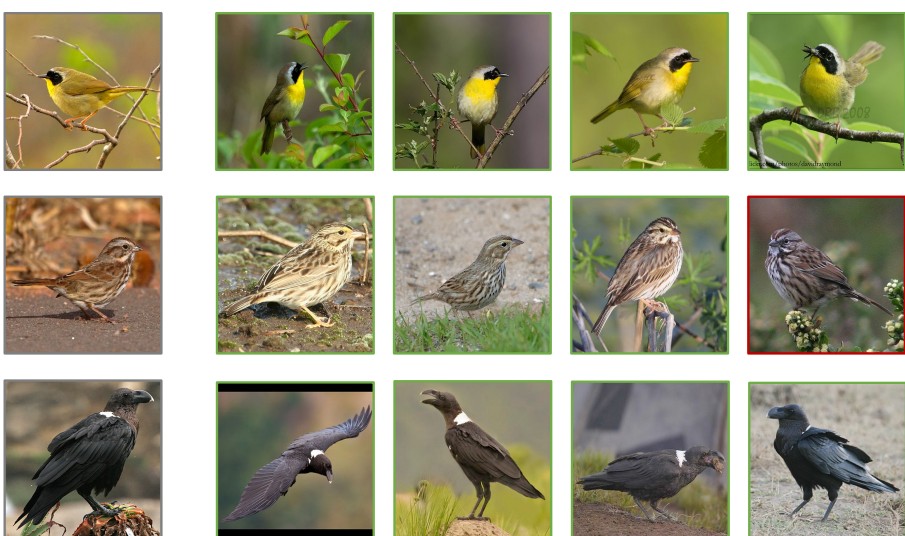

Figure 1: Qualitative results on the CUB-200-2011 dataset [1]. We present the top 4 tretrievals for each query image. The retrievals with green boundary are positive cases, and those with red boundary are failure cases. Even for failed retrievals, substantially similar visual appearances are shared with queries.

## B    Pseudo Code

We summarize the overall procedure of our MetricFormer in Algorithm 1 (Here, we set the number of batch-wise correlation blocks $T = 1$). Note that, the batch-wise correlation block and feature-wise correlation block share the same architecture.

**Algorithm 1** Pseudo Code of MetricFormer in a PyTorch-like style.

```
# X: input images for a mini-batch with B samples
# Backbone: to get the feature
# Encoder: an encoder block of Transformer
# FDB: Feature Decoupling block
# BCB: Batch-wise Correlation block
# FCB: Feature-wise Correlation block
# k: the number of sub-features
# d: the dimension of sub-features
# batch_first: a parameter in multi-head self-attention module
BCB = Encoder(batch_first=False)
FCB = Encoder(batch_first=True)
def MetricFormer(x, Backbone, FD, BCB, FCB, Training=False)
    fea_map = Backbone(X) # B×D
    fea = FDB(fea_map) # B×k×d
    if Training:
        fea = BCB(fea) # B×k×d
    fea = FCB(fea) # B×K×d
    return fea
```

## References

[1] Catherine Wah, Steve Branson, Peter Welinder, Pietro Perona, and Serge Belongie. The caltech-ucsd birds-200-2011 dataset. 2011.