# OpenReview forum: "MetricFormer: A Unified Perspective of Correlation Exploring in Similarity Learning"
_NeurIPS.cc/2022/Conference — NeurIPS 2022 Accept_

### Official Review · Reviewer_TNVU · 2022-07-09

**Rating:** 6
**Confidence:** 3
**Soundness:** 3 good
**Presentation:** 2 fair
**Contribution:** 3 good

**Summary:**

This paper proposes a method called MetricFormer, which can extract an ensemble of distinct features, model the relationship among samples in a minibatch, and discover the structural pattern in the ensemble of features. The experimental results on three public datasets are better compared with other approaches.

**Questions:**

Please see my questions in the weakness section.

**Ethics Review Area:**

["I don’t know"]

**Limitations:**

The equations and variables need to be well defined and add more explanations.

**Strengths And Weaknesses:**

Strengths:
1. The Metric transformer includes three modules: (1) feature decoupling module:  extracting an ensemble of district features; (2) batch-wise correlation block: modeling the sample relationship in each minibatch; (3) feature-wise correlation block: discovering the structural relations among the ensemble of features.
2. The proposed method results are promising in three public datasets, compared with other approaches. An ablation study is performed to show the effectiveness of each module.

Weaknesses:

The presentation of this paper needs to be improved. In equation (3), there is no relation between the two equations. Y^{-}_ should be included in the second equation.   Is the dimension or variable D at line 128 the same as the variable C at line 122? What do \alpha and \beta mean in equation (9)? What do MetricFormer-M and MetricFormer-PA mean in Table 2?

---

> ### Author Response · Authors · 2022-08-02
> **Response to Reviewer TNVU**
>
> We sincerely thank you for your time and efforts. We think all of your concerns can be addressed with the following clarifications.
>
> Q: The equations and variables need to be well defined and add more explanations.
>
> A: We will recheck the presentation and add more explanations in the final version.
> - the dimension  D on line 128 is not the same as the variable C on line 122.
> - $\alpha$ and $\beta_0$ in Eq.(9) denote two hyperparameters. In the experiment, we set $\alpha = 2$ and $\beta_0 = 1$.
> - MetricFormer-M and MetricFormer-PA denote MetricFormer combined with Margin loss and ProxyAnchor loss, respectively.

---

### Official Review · Reviewer_3kwp · 2022-07-11

**Rating:** 5
**Confidence:** 4
**Soundness:** 3 good
**Presentation:** 3 good
**Contribution:** 3 good

**Summary:**

This paper proposes a Transformer-based method named MetricFormer for metric learning. The proposed MetricFormer includes three blocks: feature decoupling block, feature-wise correlation block, and batch-wise correlation block. Given an image, feature decoupling block uses several trainable queries and cross-attention to extract different sub-features to focus on different spatial regions. Feature-wise correlation block uses self-attention to model the relations among different sub-features of a sample, while batch-wise correlation block uses self-attention to model the relations among different samples in a mini-batch.

**Questions:**

As pointed out in the "Weaknesses" part, what is the most novel point in the proposed method?

**Limitations:**

The authors have pointed out the limitation and negative impact at the end of the paper, but not provided strategies to address them. I have no suggestions for improvement to the authors either.

**Strengths And Weaknesses:**

Strengths:
- The authors conduct sufficient experiments to validate each component of their method and the results on three benchmarks achieve competitive results compared with state-of-the-art methods.

Weaknesses:
- The biggest weakness lies in the novelty. Specifically, feature decoupling block and diversity loss are consistent with [1]; feature-wise correlation block is similar to [1,2,3,4]; batch-wise correlation block is similar to [5];  Consistency Preserving Strategy is similar to [6]. All these methods have been verified in person re-id. The authors just combine them in a framework, so the novelty is limited.
- The presentation should be improved greatly. For example, it is preferred to use consistent format to list the references.

[1] Diverse Part Discovery: Occluded Person Re-Identification With Part-Aware Transformer. In CVPR, 2021.

[2] Spatial-temporal graph convolutional network for video-based person re-identification. In CVPR, 2020.

[3] Learning multi-granular hypergraphs for video-based person re-identification. In CVPR, 2020.

[4] Multi-granularity reference-aided attentive feature aggregation for video-based person reidentification. In CVPR, 2020.

[5] Spectral Feature Transformation for Person Re-Identification. In ICCV, 2019.

[6] Masked Graph Attention Network for Person Re-identification. In CVPR workshop, 2019.

---

> ### Author Response · Authors · 2022-08-02
> **Response to Reviewer 3kwp**
>
> We sincerely thank you for your time and efforts. We think all of your concerns can be addressed with the following clarifications.
>
> Q1: What is the most novel point in the proposed method?
>
> A: The core novel point of our paper is to capture multiple relationships among data and features in a unified perspective by using the designed transformer blocks.
> - As discussed in Section 1, many methods including [1-6] have explored how to excavate informative relationships among different samples and features, but no one can simultaneously capture these correlations in a unified framework. Therefore, we propose a new transformer-based method to tackle this problem. The experiments have demonstrated that our method can significantly improve performance by capturing multiple correlations.
> - Though [1-6] have discussed how to excavate informative relationships among different samples and features, the concrete techniques are different from our method.
> - The existing methods including [1-6] cannot be simply combined in a unified framework to simultaneously capture multiple correlations. Therefore, our method is not a simple combination of [1-6]. In fact, we propose a new and effective unified framework that can cleverly combine different well-designed transformer blocks to capture multiple correlations simultaneously.
> - Our method is a general similarity learning method that can be widely used in many tasks including person re-ID. [1-6] are all specific methods designed for person re-ID. Compared with them, our method has good generalization ability and is easy to be adopted in other downstream tasks.
>
> Q2: The presentation should be improved greatly.
>
> A: We will recheck and improve the presentation in the final version.

---

### Official Review · Reviewer_3PRC · 2022-07-12

**Rating:** 7
**Confidence:** 4
**Soundness:** 3 good
**Presentation:** 3 good
**Contribution:** 3 good

**Summary:**

The authors improved DML by exploring multiple informative relationships among different samples and features from a unified perspective. Current DML methods can only excavate the multiple correlations in different aspects. Considering this problem, they proposed an elaborate metric transformer named MetricFormer to effectively capture and model the multiple correlations. The proposed is generic, effective, and can be easily incorporated into many existing DML methods.

**Questions:**

- More descriptions should be added to the figures in this paper. For example, The number N of the used Batch-wise Correlation Block should be highlighted in Figure 2.

**Limitations:**

Yes, limitations have been discussed. I do not find the potential negative societal impact of this work.

**Strengths And Weaknesses:**

[Strengths]
- This is a well-written paper. The motivation is clear and the method is easy to follow. By introducing the well-designed transformer blocks, the proposed method can effectively learn more representative features.

- The authors studied DML from a new and unified perspective of correlation exploration. Previous works excavate the multiple correlations in different aspects, while the authors consider the multiple correlations from a unified perspective by the proposed MetricFormer.  The proposed method is generic. It can be widely applied to strengthen existing DML approaches since it can excavate multiple correlations in a unified perspective.

- Comprehensive experiments are provided to understand and evaluate the proposed methods.

[Weaknesses]
- Although the authors mentioned in their limitations, I think they still should report and analyze their model size and inference cost in their experimental results.

- The effectiveness of the Local Embedding Graph should be discussed in the ablation study.

---

> ### Author Response · Authors · 2022-08-02
> **Response to Reviewer 3PRC**
>
> We sincerely thank you for your time and efforts. We think all of your concerns can be addressed with the following clarifications.
>
> Q1: Although the authors mentioned in their limitations, I think they still should report and analyze their model size and inference cost in their experimental results.
>
> A: Compared with linear projection used in conventional DML methods whose computational complexity is $O(n^2d)$, the computational complexity of our transformer-based correlation computation is $O(n^2d + nd^2)$. However, recent DML methods also introduce new techniques such as GAN and GNN into the DML model, increasing computational complexity. More details will be added in the supplementary.
>
> Q2: The effectiveness of the Local Embedding Graph should be discussed in the ablation study.
>
> A: Due to the space limitation, we will add them in the supplementary.
>
> Q3: More descriptions should be added to the figures in this paper.
>
> A: We will recheck the presentation and add more descriptions in the final version.

---

> > ### Comment · Reviewer_3PRC · 2022-08-08
> > **Thanks for the response**
> >
> > Thanks for the authors’ response. The authors successfully tackle my main concerns, so I would like to vote for its acceptance.

---

### Official Review · Reviewer_dg3A · 2022-07-13

**Rating:** 7
**Confidence:** 4
**Soundness:** 3 good
**Presentation:** 3 good
**Contribution:** 3 good

**Summary:**

Towards the issue that existing methods try to excavate the multiple correlations in different aspects, this paper provides a unified perspective to learn the multiple correlations, which enables the information propagation along features of different samples within each mini-batch. The proposed MetricFormer introduces a correlation-aware transformer into the internal structure of deep metric networks. It can be ﬂexibly integrated with the existing metric learning framework. Experiments on three widely-used datasets demonstrate its superiority.

**Questions:**

Please refer to my detailed comments above.

**Limitations:**

Yes

**Strengths And Weaknesses:**

Pros:

(1) The idea of transformer-based multiple correlations exploration is interesting.

(2) The method is simple and effective. Experimental results are good.

Cons:

(1) The writing could be improved. There are several grammar mistakes and typos. For example, "a elaborate" should be "an elaborate" in line 6. The order of "Samples" and "Classes" are reversed in Table 1. The abbreviation of "PA" and "M" should be explained in Table 2.

(2) The novelty might be insufficient. The contribution of existing correlations combination  is not significant.

(3) Please analyze the computational complexity of the proposed transformer based correlation computation and compare it with existing methods.

(4) The reference is inadaquate. [1] also investigates multiple correlations for different tasks. Please also discuss the difference.

(5) Will the codes be released?

[1] Deep Comprehensive Correlation Mining for Image Clustering, ICCV 2019

---

> ### Author Response · Authors · 2022-08-02
> **Response to Reviewer dg3A**
>
> We sincerely thank you for your time and efforts. We think all of your concerns can be addressed with the following clarifications.
>
> Q1: The writing could be improved.
>
> A: We will recheck and improve the presentation in the final version.
>
> Q2: The novelty might be insufficient. The contribution of the existing correlations combination is not significant.
>
> A: The core novel point of our paper is to capture multiple relationships among data and features in a unified perspective by using the designed transformer blocks. As discussed in Section 1, many methods have explored how to excavate informative relationships among different samples and features, but no one can simultaneously capture these correlations in a unified framework. Meanwhile, the existing methods cannot be simply combined in a unified framework to simultaneously capture multiple correlations.  Therefore, we propose a new and effective unified framework that can cleverly combine different well-designed transformer blocks to capture multiple correlations simultaneously. The experiments have demonstrated that our method can significantly improve performance by capturing multiple correlations.
>
> Q3: Please analyze the computational complexity of the proposed transformer-based correlation computation and compare it with existing methods.
>
> A: We have discussed in Limitation, compared with conventional DML methods, due to the usage of the transformer, the parameters of the model are increased, which leads to more computation cost and training times. Compared with linear projection used in conventional DML methods whose computational complexity is $O(n^2d)$, the computational complexity of our transformer-based correlation computation is $O(n^2d + nd^2)$. However, recent DML methods also introduce new techniques such as GAN and GNN into the DML model, increasing computational complexity.
>
> Q4: The reference is inadequate. [1] also investigates multiple correlations for different tasks. Please also discuss the difference.
>
> A: [1] is a good work, we will add this reference in the final version. Our method is totally different from [1]: (1) The supervision is different: our method is a supervised method while [1] is an unsupervised method; (2) The field is different: our method is a similarity learning method which can be widely used in many downstream tasks, while [1] is a specific method for clustering task; (3) The correlations are different: our method excavates the pairwise correlation, batch-wise correlation, and feature-wise correlation, while [1] consider the local robustness, feature correspondence, and inter-correlation; (4) The concrete technique is different: our method uses a well-designed transformer, while [1] propose a pseudo-graph.
>
> Q5: Will the codes be released?
>
> A: Yes. We will release the codes soon after the notification.

---

> > ### Comment · Reviewer_dg3A · 2022-08-08
> > **Thanks for the response**
> >
> > Thanks for the response. My concerns are well addressed. After reading the response and other reviewers' comments, I would like to vote for its acceptance and increase my rating.

---

### Meta-Review · Area_Chair_rVzZ · 2022-08-22

**Recommendation:** Accept
**Confidence:** Certain

**Metareview:**

This paper proposes a unified framework based on the transformer model for capturing multiple relationships among data and features in similarity or metric learning. It is an interesting direction that is worth exploring, and the special way to combine multiple transformer blocks may have more general relevance. The proposed framework can be used with many existing metric learning methods. The research contribution of this paper is significant and reports good experimental results. Nevertheless, as pointed out by several reviewers, the presentation of the paper has room for improvement. The authors are recommended to put more efforts to improve its presentation. Besides, revision should also be made to address the other comments and suggestions of the reviewers, which includes explaining the novelty of this work more clearly.


**Award:**

No

---

### Decision · Program_Chairs · 2022-09-14

Accept